# Clustering of a Health Dataset Using Diagnosis Co-Occurrences

**Adrien Wartelle** [1,2,*], **Farah Mourad-Chehade** [1] , **Farouk Yalaoui** [1], **Jan Chrusciel** [2], **David Laplanche** [2] and **Stéphane Sanchez** [2]

1  Laboratoire Informatique et Société Numérique (LIST3N), Université de Technologie de Troyes, 10000 Troyes, France; farah.chehade@utt.fr (F.M.-C.); farouk.yalaoui@utt.fr (F.Y.)
2  Information Médicale et Evaluation des Performances (IMEP), Centre Hospitalier de Troyes, 10000 Troyes, France; jan.chrusciel@ch-troyes.fr (J.C.); david.laplanche@ch-troyes.fr (D.L.); stephane.sanchez@ch-troyes.fr (S.S.)
*  Correspondence: adrien.wartelle@utt.fr; Tel.: +33-(0)6-02-05-886-21

**Featured Application: Assessing patterns of healthcare problems in a general emergency department population through multimorbidity clustering analysis.**

**Abstract:** Assessing the health profiles of populations is a crucial task to create a coherent healthcare offer. Emergency Departments (EDs) are at the core of the healthcare system and could benefit from this evaluation via an improved understanding of the healthcare needs of their population. This paper proposes a novel hierarchical agglomerative clustering algorithm based on multimorbidity analysis. The proposed approach constructs the clustering dendrogram by introducing new quality indicators based on the relative risk of co-occurrences of patient diagnoses. This algorithm enables the detection of multimorbidity patterns by merging similar patient profiles according to their common diagnoses. The multimorbidity approach has been applied to the data of the largest ED of the Aube Department (Eastern France) to cluster its patient visits. Among the 120,718 visits identified during a 24-month period, 16 clusters were identified, accounting for 94.8% of the visits, with the five most prevalent clusters representing 63.0% of them. The new quality indicators show a coherent and good clustering solution with a cluster membership of 1.81 based on a cluster compactness of 1.40 and a cluster separation of 0.77. Compared to the literature, the proposed approach is appropriate for the discovery of multimorbidity patterns and could help to develop better clustering algorithms for more diverse healthcare datasets.

**Keywords:** multimorbidity patterns; emergency medical services; cluster analysis; hierarchical agglomerative clustering; health services research



## 1. Introduction

To create a coherent territorial healthcare offer, the general patient population needs to be assessed by evaluation and quantification of their health profiles. This task requires an epidemiologic analysis of the population, based on their demographic characteristics, their health problems, and their healthcare usage [1]. This evaluation could help inform public health policy, particularly in Emergency Departments (EDs), which represent a key element of the healthcare system with a role of safety net for the surrounding population. EDs are systems under tension, and admission volumes are steadily rising, with an increase from 10 to 20 million visits annually in France, corresponding to an average growth of 3.5% per year [2]. This situation has led to overcrowding and saturation of EDs, with negative repercussions on quality of care [3–5], and on working conditions for healthcare professionals. In this context, only a detailed study of healthcare needs, including analysis of patient flow and patient expectations, will enable the development of adequate solutions for healthcare delivery.

The assessment of a patient population can be done through a classification based on their sociodemographic characteristics, their healthcare usage, and a clinical evaluation of

their health state [1,6]. Particularly, in EDs, multiple classification systems exist to describe patients according to the severity of their condition and their treatment requirements [7,8]. However, they are ad hoc systems that are based on expert knowledge and not on actual data analysis. Furthermore, they are essentially targeted at allocating adequate nursing staff and not designed for detailed analysis of the population. To capture the nature of patients' health problems, the diagnosis seems to be the most appropriate candidate. It offers an adequate clinical typology of patients' visits, and a similarity structure based on either concept hierarchies [9] or co-occurrence associations [10,11]. An adequate clustering strategy using this variable would yield an unsupervised classification based on patients' recorded data with the goal of obtaining compact and well-separated cluster classes, i.e., where clustered patients are very similar to the patients of their own cluster and very dissimilar to those of other clusters [12]. Although some approaches to ED classification based on diagnoses already exist [13,14], no approach has been developed, to the best of our knowledge, using co-occurrences of diagnoses in ED patients, with multimorbidity pattern analysis.

The analysis of multimorbidity patterns constitutes an effective and appropriate tool to model and assess the complexity of patients in terms of the diversity and statistical co-occurrences of their health conditions [15]. The literature on multimorbidity patterns was reviewed in 2014 by Pradros-Torres et al. and in 2019 by Busija et al., and several methods and population contexts were identified [16,17]. The methods were factor analysis, to discover underlying common factors of diseases [18,19]; clustering analysis, to regroup either diseases or people into clusters [20–22]; or network analysis, to study the links between pairs of diseases [23]. These approaches are mainly focused on disease analysis of complex and chronic patients, which represent the greatest burden on the healthcare economy. Many opportunities still exist to extend the application of theses methodologies to broader contexts, across wider populations and with more robust strategies. Therefore, the goal of this study was to implement a new methodology to analyze multimorbidity patterns in an ED population.

This paper proposes an original hierarchical clustering approach, based on multi-morbidity analysis, for the classification of ED patients' visits. The proposed approach creates clusters by merging patient profiles according to their diagnosis co-occurrences. Having a patient dataset, it uses patient diagnoses from the International Classification of Diseases 10th revision (ICD10) and measures their similarities by computing a relative risk (RR) of co-occurrence. Then, the algorithm constructs a clustering dendrogram solution by iteratively agglomerating clusters, beginning with singleton clusters and finishing with a unique global cluster, using a hierarchical strategy. However, the proposed method differs from the classical Hierarchical Agglomerative Clustering (HAC) algorithms [20,24] as it does not use any linkage strategy to compute similarities at cluster-level, but uses only its present RR measure. Furthermore, it introduces relevant cluster quality indicators related to cluster compactness, cluster separation and cluster membership. The RR measure is therefore computed at each agglomeration to compare clusters, to compute quality indicators and to construct a rule for selecting the best cluster-pair to merge at each agglomeration. The ultimate goal of this strategy is to construct a clustering solution with maximal cluster membership and sufficient cluster compactness (higher than 1) by selecting the best proposal from the set produced by the agglomerations. The proposed approach has been applied to the dataset of the largest ED in the Aube Department (Eastern France). This new clustering design performs better than state-of-the-art methods, by allowing the construction of a coherent and clinically meaningful clustering solution.

The rest of the paper is organized as follows. Section 2 presents the materials and methods with the ED health record dataset involved and the new multimorbidity algorithm based on HAC. Section 3 presents the results, with a comparison of several clustering algorithms and a description of the best clustering solution obtained though the new quality indicators and the characteristics of the cluster visits. Section 4 presents a discussion comparing the method and the results of the original method with traditional approaches



at three stages of the clustering algorithm design. Section 5 presents the conclusion of the study, placing our algorithm in the general context of multimorbidity pattern analysis and opens new perspectives.

## 2. Materials and Methods

This section presents the study context with the characteristics of the ED health record dataset used for analysis. It describes the relevant variables used for the description of clusters and for their construction with the diagnoses that are at the core of the new algorithm. Finally, and above all, it presents the original multimorbidity clustering strategy using a specific relative risk of co-occurrences as a similarity measure, new quality indicators and an algorithm presenting a pertinent agglomerative strategy with adequate rules.

### 2.1. Study Design, Setting and Population

The population under study in this paper is the whole population of patients attending the ED of Troyes hospital from 1 May 2017 to 28 April 2019. Troyes hospital is the largest hospital in the Aube Department of Eastern France, with a population of 310,000 inhabitants and a medical density of 234.1 physicians per 100,000 inhabitants, placing it in the lowest quarter of all the Departments in France. The hospital has 442 medical beds, 127 surgical beds, and 63 beds dedicated to gynecology/obstetrics. In 2018, there were a total of 62,082 ED visits, and an average rate of use of 250 to 330 visits per 1000 inhabitants within the hospital's catchment area. With annual ED visits higher than 45,000, according to national statistics [25], the ED of Troyes hospital is classed as having a very high volume of activity. During the 2-year study period, 120,718 visits to the ED were recorded, involving 75,279 patients; 114,391 of these visits (94.76%), involving 72,666 patients (96.53%), are retained for the present analysis based on the presence of at least one diagnosis. A total of 6199 visits (5.16%), involving 5459 patients (7.25%), were excluded from the clustering analysis due to missing diagnostic codes, and a further 208 (0.17%) because of an insufficient number of diagnosis occurrences, defined here as a minimum of 10 to minimize sampling bias. The study focuses on the general population without stratification, to keep the assessment as general as possible.

ED visits are described according to the characteristics of the patient, i.e., their age, gender, visit frequency, visit prescription, triage circuit followed at arrival, hospital admission and visit diagnosis. The triage circuit can be either the long circuit (normal evaluation and management), or the short (fast-track) circuit, which is a pathway for assessment and treatment of low-severity patients (usually minor injuries or benign medical conditions). Thus, the characteristics of the 114,391 ED visits can be described with a mean age of 39.84 years (with a large standard deviation of 27.40 years), 50.86% females, a mean number of visits per patient of 1.57, a mean number of medications per visit of 1.45 (with a large standard deviation of 2.87), a probability of biological exams of 42.9%, of radiological exams of 31.7%, a ratio of short circuit to long circuit of 1.81, and a 22% hospitalization rate.

The diagnosis, used here as a central active variable for clustering, is represented using the International Classification of Diseases 10th revision (ICD10), which is a classification documented by the World Health Organization (WHO) that uses an alphanumerical representation with a letter followed by a series of 2 to 6 numbers depending on the level of precision of the diagnosis. Its hierarchical representation can be divided into 22 chapters (accounting) with blocks of categories which are subsets of chapters, categories of 3 characters, sub-categories of 4 and 5 characters, and complete codes [26]. For example, the block A00-A09 is a block from the chapter A00-B99 *Certain infectious and parasitic diseases* and it contains *Intestinal infectious diseases* diagnoses, which begin with A0 followed by a number between 0 and 9 and any number series after that (of 0 to 4 numbers). The number of existing ICD-10 codes at several hierarchical levels is not well referenced as the manual provided by the WHO does not indicate them [27]. The number of complete codes is above 155,000 [26], the number of categories is limited to 2600, and the number of block of categories varies based on how the categories are grouped, but a possible set with 263 blocks

has already been used in the literature [21]. For the ED dataset used in this 2-year study, 2890 complete codes were found, with 1017 categories and 162 blocks of categories. An appropriate description level of the diagnosis is required to ensure adequate statistical power with enough occurrences of each diagnostic label, while enabling precise identification of the health problem. Dynamic lexical representation according to a population threshold can be used for this purpose [28]. Given our population size, the general purpose of the study, and to maintain coherent representation through a similar hierarchical concept level, blocks of categories were chosen as a description level with 151 labels identified as having more than 10 occurrences on episodes of visits (see Supplementary Table S1).

### 2.2. Statistical Analysis: A New Hierarchical Agglomerative Clustering Algorithm

The proposed clustering algorithm is based on the analysis of multimorbidity patterns to unveil diagnosis relationships. From the 151 recorded blocks of ICD-10 diagnoses, it produces hard diagnosis clusters, which are then used to classify the ED visits given their recorded diagnoses. The diagnosis gives a precise and concise expert opinion on the health problem justifying the visit and, therefore, it is the most clinically informative and pertinent variable. It was the only active variable as the use of other active variables like age, gender or prescription data would have introduced unnecessary complexity in the algorithm and in the interpretation of results. This is due to the specialized approaches implemented by multimorbidity clustering which focus on the diagnosis variable. Selection or stratification strategies can be used if specific populations are of interest. Here, we avoided this strategy as the focus is on a general population assessment. Stratification would have made the results harder to evaluate, because of the redundant patterns between strata that it would have produced. Contrary to clustering based directly on patients or visits, where distance measure is often required, this approach proposes a simple solution for the classification of ED visits through the recorded diagnoses. Multimorbidity is defined here by the occurrence of two ICD-10 diagnostic codes, from the 151 blocks of categories, in one episode of ED visits. An episode of ED visits is defined for each patient as a set of consecutive ED visits separated by a maximum of 6 months, to limit non-pertinent and "noisy" associations. Using these co-occurrences to gather the initial blocks through the new clustering algorithm, clusters of diagnoses are identified, characterized, and used for the classification of ED visits. In the following sections, we first introduce the similarity measure based on multimorbidity. We then present the new quality indicators and the original clustering algorithm.

### 2.2.1. Similarity Measure

Clustering categorical data like ICD10 diagnostic codes is challenging, as the lack of direct continuous and Euclidian distance-based similarity measures requires innovative approaches [29]. In this study, we propose to take advantage of multimorbidity, relying on statistical co-occurrences of pairs of diagnoses on patterns, to measure their similarity. Hence, we introduce a new measure of similarity based on the risk of co-occurring diagnoses in an episode of ED visits, namely the relative risk described by Equation (1):

$$RR_{ij} = \frac{p_{ij}}{p_i p_j} \tag{1}$$

where $p_{ij}$ designates the probability of diagnoses i and j co-occurring in episodes of ED visits and $p_i$ (respectively $p_j$) is the marginal probability that block i (respectively j) occurs in an episode and is used to weight the relation [30]. These probabilities could be computed using raw empirical observations from the contingency tables, which contain co-occurrences of pairs of diagnoses. Let $a_{ij}$, $b_{ij}$, $c_{ij}$ and $d_{ij}$ be the numbers of visits verifying, respectively, the positive-positive, positive-negative, negative-positive and negative-negative matches in

the contingency table related to diagnostics i and j. Then, $p_{ij} = \frac{a_{ij}}{N_e}$, $p_i = \frac{a_{ij} + b_{ij}}{N_e}$, $p_j = \frac{a_{ij} + c_{ij}}{N_e}$ and thus, Equation (1) is equivalent to Equation (2):

$$RR_{ij} = \frac{a_{ij} N_e}{(a_{ij} + b_{ij})(a_{ij} + c_{ij})} \tag{2}$$

with $N_e = a_{ij} + b_{ij} + c_{ij} + d_{ij}$ the total number of episodes, independent of the considered diagnoses. Even though this comorbidity measure is not new [31], this study is the first to use it for multimorbidity clustering purposes. This measure was chosen because it focusses on the relevant positive matches $a_{ij}$ and provides a comprehensive interpretation with a comparison between the observed co-occurrence probability $p_{ij}$ and the probability $p_i p_j$, which corresponds to the probability of co-occurrence in the case where diagnoses would have been probabilistically independent. The relative risk measure $RR_{ij}$ will be used for similarity computations, instead of the Jaccard similarity measure $J_{ij}$ usually used with HAC approaches in the literature [16,17,20]. In our study, $J_{ij}$ would be equivalent to $\frac{p_{ij}}{p_i + p_j - p_{ij}} = \frac{a_{ij}}{a_{ij} + b_{ij} + c_{ij}}$. Compared to our relative risk measure, the Jaccard similarity measure is less sensitive to variation in marginal probabilities and does not consider well the external scarcity context of marginal occurrences that allow a better appreciation of the co-occurrence's strength. Furthermore, it cannot be associated with the notion of probability independence. Yule's Q coefficient of association, equivalent to $\frac{a_{ij} d_{ij} - b_{ij} c_{ij}}{a_{ij} d_{ij} + b_{ij} c_{ij}}$, is also frequently used as a similarity measure, but is not focused on positive matches $a_{ij}$ [32].

### 2.2.2. New Quality Indicators

The relative risk measure introduced in the previous section is at the foundation of all measures of the clustering algorithm. Indeed, it is used to construct clustering quality indicators, namely $RR_{intra}$, which measures cluster intra-compactness by computing the average of the relative risk of a block with the other blocks of its cluster, and $RR_{inter}$, which measures cluster inter-separability by computing the average of the maximum relative risk of a block with a cluster other than its own. Let $S = \{C_1, \ldots, C_N\}$ be some clustering solution composed of a set of N clusters denoted $C_k$, $k \in \{1, \ldots, N\}$, each cluster having a certain number of diagnosis blocks. The relative risk $RR_{i,C_k}$ of a diagnosis i co-occurring with a cluster $C_k$ is computed directly with the Equation (1) by considering the cluster $C_k$ as a new diagnosis labelled j in the dataset and by taking all the visits of any diagnosis of $C_k$ in a single pool of visits related to $C_k$.

Then, the cluster intra-compactness and inter-separability indicators for a clustering solution S can be formulated as follows:

$$RR_{intra}(S) = \sum_{\substack{i \in D \\ C_k \in S \,|\, i \in C_k}} p_i{}' * RR_{i,(C_k \setminus \{i\})} \tag{3}$$

$$RR_{inter}(S) = \sum_{i \in D} p_i{}' * \max_{C_k \in S \,|\, i \notin C_k} RR_{i,C_k} \tag{4}$$

where D is the set of diagnosis blocks, i is the index of some diagnosis block, $p_i{}' = \frac{p_i}{\sum_{i \in D} p_i}$ is the empirical probability that a block occurrence on ED episode is from block i and $C_k \setminus \{i\}$ is the cluster k with diagnosis block i excluded. The indicator $RR_{intra}(S)$ is designed to ignore co-occurrences of a diagnosis with itself, as it does not yield any pertinent information; it would correspond to the inverse of its occurrence probability $\frac{1}{p_i}$ and therefore $RR_{intra}(S)$ would depend much more on the distribution of block occurrence probabilities than on the distribution of their co-occurrence probabilities. However, this design implies an ad hoc choice for singleton clusters. Indeed, when a block of diagnosis i is a singleton, we define $RR_{i\varnothing} \equiv 1$ as it provides an adequate reference value corresponding to probability independence. The indicator $RR_{inter}(S)$ is designed to use a maximum RR value instead of

an average RR value with all other clusters (other than the one of the blocks involved) for two main reasons. The first is that most blocks are related to only a few other blocks and thus to only a few other clusters and an average RR value with all other clusters would mitigate the measure of pertinent relationships and would systematically return a low value. The second reason is that it provides an indication, at a block level, of what would be the best alternative cluster that each block could be classified in. The relative strength of this alternative link can be given by comparing $RR_{intra}(S)$ and $RR_{inter}(S)$ values at a block level, thus making the two quality indicators comparable.

The two new quality indicators introduce new concepts for analyzing multimorbidity patterns. To the best of our knowledge, they are the first specific and relevant clustering indicators dedicated to this purpose. Furthermore, as their comparison is meaningful, we can introduce a global indicator for the clustering algorithm, namely the Membership Ratio (MR) criterion:

$$MR(S) = \frac{RR_{intra}(S)}{RR_{inter}(S)} \tag{5}$$

This global clustering quality indicator is constructed in the same spirit as most classical ones like the Dunn index [33], by comparing an indicator of intra-quality of clusters to an indicator of inter-quality of clusters. The goal of the clustering is to construct a clustering solution $S^*$ that will maximize this indicator, $S^* = \text{argmax}_S MR(S)$. However, the MR indicator has the particularity of being non-distance-based, and therefore it requires an adapted approach to be optimized.

### 2.2.3. The HAC-Cooc Algorithm

This section describes the proposed Hierarchical Agglomerative Clustering for co-occurrence analysis, denoted HAC-cooc, which clusters the blocks of diagnoses [23,24]. The algorithm constructs a clustering solution by first considering each diagnosis block as a cluster and then by iteratively agglomerating pairs of diagnosis clusters using a local rule based on the maximization of the RR measure. Let $S^{(t)} = \left\{ C_1^{(t)}, \ldots, C_{N^{(t)}}^{(t)} \right\}$ be the clustering solution at iteration $t \in \{1, 2, \ldots, D-1\}$ of the algorithm. Instead of using the Lance and Williams recurrence formula traditionally used to update the measures of similarity with each new agglomeration [34], the relation (1) is performed at each stage of agglomeration for each pair of clusters, leading to a set of $RR_{C_k^{(t)}, C_l^{(t)}}$, $\forall k, l \in \left\{ 1, \ldots, N^{(t)} \right\}$. To select the cluster pair to agglomerate, two rules (a) and (b) are used and compared. The rule (a) can be expressed as:

$$\theta_a^{(t)} = \text{argmax}_{k,l \in \{1,\ldots,N^{(t)}\}} RR_{C_k^{(t)}, C_l^{(t)}} \tag{6}$$

where $\theta_a^{(t)}$ is the pair of clusters indices chosen for current agglomeration with rule (a). Rule (a) simply selects the best current link while rule (b) uses a more indirect approach:

$$\theta_b^{(t)} = \text{argmax}_{k,l \in \{1,\ldots,N^{(t)}\}} \frac{RR_{C_k^{(t)}, C_l^{(t)}}}{\max\left( \min\left( \max_{1 \leq l' \leq N^{(t)} | l \neq l'} RR_{C_k^{(t)}, C_{l'}^{(t)}}, \max_{1 \leq k' \leq N^{(t)} | k \neq k'} RR_{C_{k'}^{(t)}, C_l^{(t)}} \right), 1 \right)} \tag{7}$$

where $\theta_b^{(t)}$ is the pair of clusters indices chosen for current agglomeration with rule (b). Rule (b) is designed to select the most exclusive link by comparing the $RR_{C_k^{(t)}, C_l^{(t)}}$ measure of a pair of clusters $C_k^{(t)}$ and $C_l^{(t)}$ with the other RR measures of clusters k and l. This method was hypothesized to be safer and better in the long run than choosing local optimums, by considering the multiple relations that each diagnosis cluster has with the others.

Applying either rule (a) or (b) leads to a new clustering solution, having one less cluster than the previous one. The algorithm continues the agglomerations until all blocks are distributed into two clusters, the solution with one cluster being irrelevant. This

process produces a set of clustering solutions, each obtained at a stage of agglomeration. The best solution $S^*$, with a particular number of clusters, is determined by maximizing the global quality criterion MR, $S^* = \text{argmax}_{S^{(t)}} MR\left(S^{(t)}\right)$. This solution represents the optimal partition of diagnosis blocks within a set of optimal clusters. The global constraint $RR_{intra} > 1$ is imposed on the clustering solution to obtain a solution where clusters are compact, meaning here that the co-occurrence probability is on average higher than would be expected by chance. Algorithm 1 describes the approach of the HAC-cooc algorithm with the initialization phase, the agglomeration routine phase and the selection of the optimal solution phase (Supplementary Algorithm S1 illustrates a possible implementation of algorithm 1 using matrices).

---

**Algorithm 1** The new Hierarchical Agglomerative Clustering (HAC-cooc) algorithm pseudo-code.

---

```
01.  //Initialization
02.  D: Total number of diagnoses to be clustered
```
03. $N^{(1)} = D$:  Initial total number of clusters
04. $C_k^{(1)}$, $k \in \{1,\ldots,D\}$:  Initial $k^{th}$ singleton cluster, with the diagnosis k
05. $S^{(1)} = \left\{C_1^{(1)}, \ldots, C_D^{(1)}\right\}$:  First clustering solution
06. $I = \varnothing$:  Indices set of possible clustering solutions
```
07.
08.  //Agglomeration Routine
```
09. For$(t = 2,\ldots, D-1)$ {
10.    Consider each pair of clusters $\left(C_k^{(t-1)}, C_l^{(t-1)}\right)$, $k,l \in \left\{1,\ldots,N^{(t-1)}\right\}$
11.    Compute $RR_{C_k^{(t-1)},C_l^{(t-1)}}$ for each pair using (1)
12.    Select the best link $\theta_r^{(t-1)} = (k^*, l^*)$, with r = a or b using (6) or (7)
13.    Merge $C_{k^*}^{(t-1)}$ and $C_{l^*}^{(t-1)}$: $C_{new}^{(t)} = C_{k^*}^{(t-1)} \cup C_{l^*}^{(t-1)}$
14.    Compute the clustering solution $S^{(t)} = \left(S^{(t-1)} \backslash \left\{C_{k^*}^{(t-1)}, C_{l^*}^{(t-1)}\right\}\right) \cup C_{new}^{(t)}$
15.    Associate the visits of $C_{k^*}^{(t-1)}$ and $C_{l^*}^{(t-1)}$ to $C_{new}^{(t)}$
16.    Compute $N^{(t)} = N^{(t-1)} - 1$
17.    Compute the quality indicators $RR_{intra}\left(S^{(t)}\right)$ and $RR_{inter}\left(S^{(t)}\right)$ using (3) and (4)
18.    Compute the quality indicator $MS\left(S^{(t)}\right)$ using (5)
19.    If $RR_{intra}\left(S^{(t)}\right) > 1$ {
20.        $I = I \cup \{t\}$
21.    }
22. }
23.
```
24.  //Selection of the optimal solution
```
25. $S^* = \text{argmax}_{S^{(t)}, t \in I} MR\left(S^{(t)}\right)$

---

The complexity of Algorithm 1 is driven by the operation at lines 10, 11, 12 and 13, which each have a complexity $O\left(N^{(t-1)}\right)$ or $O\left(\left(N^{(t-1)}\right)^2\right)$ depending on whether they are optimized or not, i.e., whether the algorithm only updates proximity measures that have changed, or if it computes them for all pairs at each loop. It is also driven by the operation at line 15, which has a complexity of $O(N_v)$ as it changes the association between the $N_v$ visits and the clusters. Therefore, considering the loop on t, the algorithm complexity is quadratic with $O(\max(D, N_v) * D)$ with the optimized form and third order with $O\left(\max\left(D^2, N_v\right) * D\right)$ otherwise. Using a non-optimized (third order) implementation with the R language (v4.0.1) and an Intel i5-4300M CPU at 2.60 GHz, the execution time was of 17 min with D = 151 and $N_v = 114\,391$. As $N_v > D^2$, the complexity is driven linearly by the number of visits ($O(N_v * D)$) and the optimized version does not offer a reduction in complexity. A sensibility analysis was performed on a nine-week period with D = 96 and $N_v = 9317$ resulting in execution time of 2 min and 30 s, which is coherent

with the complexity analysis. Keeping this configuration with $D$ constant and considering the linear relationship with $N_v$, a dataset with 1 million visits ($N_v$) would imply a run time between 3 and 5 h and one with 10 million visits ($N_v$) around 2 days, which begins to be quite long but still acceptable. However, with more visits, it would also be interesting to use more diagnoses that are more precise and thus increase D by scale factor between 1 and 10 for each increase in lexical depth of ICD10 codes.

## 3. Results

This section illustrates the effectiveness of the proposed clustering approach, with both rules of agglomerations (a) and (b), named, respectively, HAC-cooc Rule (a) and HAC-cooc Rule (b). We also compare our approach to four other types of approaches. The first, named HAC –RR, uses the classical HAC algorithm with −RR for dissimilarity. The second, named HAC Jaccard, uses $1 - J$ as the initial dissimilarity measure, with J being the Jaccard similarity [20]. The third, named HAC Yule, uses $1 - Q$ as the initial dissimilarity measure, with Q being Yule's Q coefficient [32]. Finally, the fourth, named MCA HAC uses Multiple Correspondence Analysis (MCA), combined with HAC applied on the embedded space produced by the MCA on the diagnosis blocks [21]. Multiple Correspondence Analysis is a generalization of Principal Component Analysis (PCA) for categorical variables and of Correspondence Analysis for multiple categorical variables. It allows embedding of categorical variables in a Euclidean space through statistical analysis of their occurrence and co-occurrence patterns using a singular value decomposition on a matrix derived from the indicator matrix of categories [35]. A sensitivity analysis was also included to evaluate the robustness of the algorithm regarding the study period.

### 3.1. Selection of the Clustering Solution

All approaches were applied to cluster the diagnoses of the health dataset of the ED and the quality indicator MR was computed. Figure 1 illustrates the approaches' HAC-cooc Rule (a), HAC-cooc Rule (b) and HAC Jaccard with the MR criterion as a function of the number of clusters. The plot shows increasing MR values with the decreasing number of clusters. While these former approaches performed well, the HAC-RR and MCA HAC approaches (for all cluster linkage) were inappropriate, because they did not identify any patterns: they produced a single cluster catching all singleton block clusters at each agglomeration. Particularly, the shared variance explained by the first two components produced by the MCA approach was only 1.87%, which is very low (compared to the minimum of 1.32%) and indicates a very poor ability to find clusters.

The HAC Jaccard approach succeeded in producing an adequate clustering distribution, but with poor performance regarding the indicator of the present study, with a mean MR of 0.496 for solutions from 5 to 100 clusters. Even though the MR was 1.53, 3.82 and 4.53 for four, three and two clusters, respectively, the $RR_{intra}$ was too low with a value of 0.47 for the three solutions. Using the Jaccard coefficient directly as the rule for the HAC-cooc approach performed even worse, with MR values between 0.1 and 0.31 with the optimum number of clusters being 103. The HAC Yule approach provided good clustering results with better quality performance for solutions between 130 and 70 clusters with an MR ranging from 0.44 to 1.47. Although it performed less well than our proposed approaches, the HAC Yule approach still yielded good quality solutions, with an MR between 0.98 and 1.53 for solutions with $RR_{intra} > 1$ with an optimal result at 47 clusters, and a maximal value of 1.58 at 2 clusters but $RR_{intra} = 0.45$.

For both the direct (rule (a)) and indirect strategies (rules (b)), the HAC-cooc approach had the best overall quality performances compared to the state-of-the art algorithms. Rule (a) was initially better for solutions with 45 to 115 clusters. However, for solutions with less than 45 clusters, rule (b) showed a sudden rise and thus rules (a) and (b) produced similar clustering solutions in terms of MR, which varied between 1.6 and 1.82 for solutions between 45 and 13 clusters. With rule (a), the best solution was with 17 clusters, an MR of 1.77 based on an $RR_{intra}$ of 1.5 and $RR_{inter}$ of 0.82. Rule (b) produced a better solution

with 16 clusters, an MR of 1.81 based on an $RR_{intra}$ of 1.40 and $RR_{inter}$ of 0.77. The solution produced by rule (b) was thus selected for analysis.

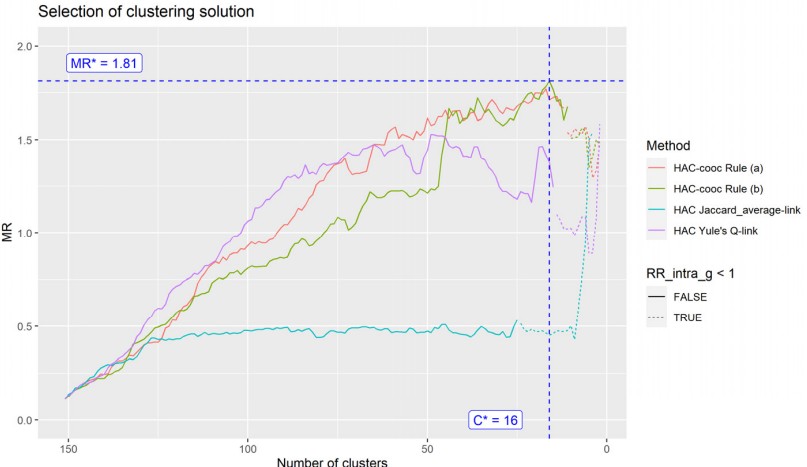

**Figure 1.** Graph of the evolution of the membership ratio (MR) criterion for each agglomeration stage of the hierarchical agglomerative clustering (HAC)-cooc Rule (a), HAC-cooc Rule (b) and HAC Jaccard approaches. Each agglomeration stage can be linked to several clusters by subtracting the number of agglomerations from the initial number of blocks of diagnoses.

### 3.2. Analysis of the Clustering Quality

This section aims to illustrate the quality of clustering by a detailed analysis of the MR indicator statistics as well as the analysis of spread of the episodes of visits over the clusters illustrated by Tables 1 and 2 and Figure 2. The quality of the clustering can be reflected by the clinical coherence of the blocks that are clustered. It enabled identification of each by a name summarizing the health problems related to each block. Furthermore, the quality of the clustering, which can be summarized by the global MR of 1.81, can also be quantified, and therefore analyzed at block and at cluster level. Table 1 shows that a good quality repartition with 115 blocks (76.7%) having an MR above 1, meaning that they are more related to the other blocks of their cluster than they are with any other cluster. Furthermore, these 115 blocks account for a vast majority of 101,527 block occurrences (94.8%) with 57.0% having a good quality with an MR between 1.0 and 2.0 and 37.7% having an excellent quality with an MR above 2.0. Table 2 indicates further that, for the block with a prevalence in cluster above 2% (see Supplementary Table S2 to see the complete classification), almost all blocks have an MR above 1. Out of the 82 blocks described in Table 2, 78 blocks verify this observation, and the four exceptions are N10-N16 in cluster 1, R25-R29 in cluster 2, S50-S59 in cluster 5 and F50-F59 in cluster 13 with MRs of 0.94, 0.92, 0.58 and 0.61, respectively. This observation is also true at the cluster level for all 16 clusters where the MR varies between 1.25 (cluster 2) and 3.42 (cluster 16). Particularly, clusters 2 and 4 from the first group, containing blocks related to general symptoms and to unspecific and broad diagnoses, have the lowest MR, at 1.25 and 1.40, respectively. In contrast, the clusters with the highest MR of 3.42 and 3.64, depicted in Figure 2, were 9 and 16 from the second group, with a very precise trauma health problem and only two blocks each.

**Table 1.** Distribution of the MR indicator for the 151 blocks.

| MR Interval. | Number of Blocks Concerned, n (%) | Number of Occurrences Concerned, n (%) |
|:---:|:---:|:---:|
| **(0.1, 0.5]** | 10 (6.6%) | 465 (0.4%) |
| **(0.5, 1.0]** | 25 (16.5%) | 5129 (4.8%) |
| **(1.0, 1.5]** | 47 (31.1%) | 34,503 (32.3%) |
| **(1.5, 2.0]** | 24 (15.9%) | 26,444 (24.7%) |
| **(2.0, 4.0]** | 36 (23.8%) | 37,616 (35.2%) |
| **(4.0, 7.4]** | 9 (6.0%) | 2964 (2.5%) |

**Table 2.** Diagnostic content and quality indicators of the 16 clusters.

| Cluster Identification N (%) MR (RR_intra/ RR_inter) | Block of Diagnoses (Prevalence >2%) : MR (RR_intra/ RR_inter) | Cluster Identification N (%) MR (RR_intra/ RR_inter) | Block of Diagnoses (Prevalence >2%) : MR (RR_intra/ RR_inter) |
|---|---|---|---|
| **1: Digestive disorders, pregnancy, menstruation** N = 17705 (15.5%) MR = 1.91 (1.48/0.77) | R10-R19 (37.1%): 2.24 (1.42/0.63) K55-K64 (16.2%): 1.09 (0.87/0.80) N80-N98 (8.7%): 3.13 (2.01/0.64) N20-N23 (6.9%): 1.06 (0.67/0.63) O00-O08 (5.3%): 5.06 (2.88/0.57) N30-N39 (4.9%): 1.22 (1.33/1.09) Z30-Z39 (4.1%): 2.85 (2.91/1.02) O20-O29 (3.8%): 3.59 (3.10/0.86) K80-K87 (3.8%): 1.34 (0.97/0.72) *N10-N16 (3.7%): 0.94 (1.00/1.07)* N70-N77 (2.1%): 1.71 (1.75/1.03) | **9: Feet Trauma** N = 5224 (4.6%) MR = 3.65 (2.18/0.60) | S90-S99 (98.5%): 3.76 (2.18/0.58) |
| **2: General symptoms and mental disorders** N = 14336 (12.5%) MR = 1.25 (1.00/0.80) | R50-R69 (68.4%): 1.33 (0.97/0.73) G40-G47 (11.1%): 1.62 (1.01/0.62) *R25-R29 (7.7%): 0.92 (1.39/1.50)* I60-I69 (6.0%): 1.16 (0.92/0.80) H80-H83 (5.2%): 1.20 (0.68/0.57) | **10: Back and Spine disorders** N = 3777 (3.3%) MR = 2.06 (1.26/0.61) | M40-M54 (79.7%): 2.31 (1.26/0.55) S30-S39 (20.4%): 1.47 (1.26/0.86) |
| **3: Infectious diseases** N = 14336 (12.5%) MR = 1.79 (1.48/0.82) | J00-J06 (21.7%): 2.95 (1.74/0.59) Z70-Z76 (13.8%): 1.37 (1.14/0.83) J40-J47 (8.6%): 1.28 (1.36/1.07) H65-H75 (7.8%): 2.07 (1.70/0.82) J20-J22 (7.4%): 2.16 (1.84/0.85) A00-A09 (7.3%): 2.36 (1.75/0.74) K50-K52 (6.1%): 1.82 (1.58/0.86) T66-T78 (4.4%): 1.21 (0.72/0.59) K20-K31 (4.2%): 1.01 (1.22/1.21) L50-L54 (3.6%): 1.80 (0.89/0.49) B00-B09 (3.4%): 1.96 (2.03/1.04) R20-R23 (3.4%): 1.18 (1.21/1.03) L20-L30 (2.2%): 1.60 (1.44/0.90) | **11: Occulomotor disorders** N = 3235 (2.8%) MR = 2.47 (1.29/0.52) | T15-T19 (37.0%): 2.35 (0.93/0.39) H10-H13 (22.6%): 1.85 (1.09/0.59) H55-H59 (15.9%): 5.08 (1.89/0.37) H53-H54 (9.0%): 2.33 (1.35/0.58) H15-H19 (6.6%): 2.71 (1.81/0.67) H00-H06 (5.3%): 3.38 (1.75/0.52) |
| **4: General symptoms of chronic conditions** N = 13893 (12.1%) MR = 1.40 (1.36/0.97) | R00-R09 (48.4%): 1.55 (1.19/0.77) J09-J18 (12.8%): 1.43 (1.28/0.90) I30-I52 (7.9%): 1.84 (1.92/1.04) E79-E90 (3.5%): 1.33 (1.34/1.00) D60-D64 (2.6%): 1.02 (1.80/1.76) D50-D53 (2.5%): 1.41 (1.67/1.19) E10-E16 (2.3%): 1.18 (1.00/0.85) I20-I25 (2.2%): 2.54 (1.41/0.56) | **12: Lower limb trauma** N = 3074 (2.7%) MR = 1.56 (1.00/0.64) | S80-S89 (100.0%): 1.56 (1.00/0.64) |
| **5: Mental disorders and at-risk behaviours** N = 12139 (10.6%) MR = 1.65 (1.51/0.91) | R40-R46 (17.9%): 1.44 (1.27/0.88) Z00-Z13 (15.8%): 1.50 (1.22/0.82) S50-S59 (15.4%): 0.58 (0.47/0.82) F10-F19 (11.0%): 1.04 (1.71/1.64) F40-F49 (10.3%): 1.74 (1.69/0.97) T36-T50 (8.6%): 3.06 (2.30/0.75) F30-F39 (7.9%): 3.82 (2.78/0.73) T51-T65 (3.4%): 1.04 (0.60/0.57) F20-F29 (3.3%): 4.54 (3.42/0.75) S10-S19 (3.3%): 1.37 (0.76/0.55) F90-F98 (2.1%): 3.24 (3.69/1.14) | **13: Cutaneous infections, wounds, and skin disorders** N = 2787 (2.4%) MR = 2.18 (1.75/0.80) | L00-L08 (38.9%): 2.79 (1.2/0.55) Z20-Z29 (22.5%): 1.55 (1.18/0.76) K00-K14 (21.1%): 2.03 (1.33/0.66) L60-L75 (4.6%): 4.79 (3.48/0.73) L80-L99 (2.7%): 1.65 (3.34/2.02) F50-F59 (2.2%): 0.61 (1.29/2.12) |

<div align="center">

**Table 2.** *Cont.*

</div>

| Cluster Identification<br>N (%)<br>MR (RR_intra/<br>RR_inter) | Block of Diagnoses (Prevalence<br>>2%) : MR (RR_intra/ RR_inter) | Cluster Identification<br>N (%)<br>MR (RR_intra/<br>RR_inter) | Block of Diagnoses (Prevalence<br>>2%) : MR (RR_intra/ RR_inter) |
|---|---|---|---|
| **6: Wrist and hand trauma**<br>**N = 6777 (5.9%)**<br>**MR = 1.75 (1.00/0.57)** | S60-S69 (100.0%): 1.75 (1.00/0.57) | **14: Arthropathies**<br>**N = 2760 (2.4%)**<br>**MR = 2.63 (1.88/0.71)** | M60-M79 (70.9%): 3.10 (1.88/0.60)<br>M00-M25 (29.2%): 1.90 (1.88/0.99) |
| **7: Head Trauma**<br>**N = 6112 (5.3%)**<br>**MR = 2.27 (1.00/0.44)** | S00-S09 (100.0%): 2.27 (1.00/0.44) | **15: Shoulder and arm trauma**<br>**N = 1930 (1.7%)**<br>**MR = 1.82 (1.00/0.55)** | S40-S49 (100.0%): 1.82 (1.00/0.55) |
| **8: Hip related trauma and disorders**<br>**N = 5781 (5.1%)**<br>**MR = 1.85 (1.90/1.03)** | S70-S79 (21.2%): 1.25 (1.19/0.96)<br>R30-R39 (14.6%): 3.11 (2.90/0.93)<br>A30-A49 (11.0%): 1.33 (1.44/1.09)<br>T20-T32 (9.7%): 1.03 (0.52/0.50)<br>N40-N51 (9.3%): 2.17 (1.48/0.68)<br>I80-I89 (7.7%): 1.08 (1.13/1.05)<br>T79-T79 (6.6%): 1.60 (1.63/1.02)<br>T80-T88 (5.8%): 3.83 (6.39/1.67)<br>I70-I79 (3.6%): 1.29 (2.04/1.57)<br>Z40-Z54 (3.3%): 2.54 (3.40/1.34)<br>D65-D69 (2.4%): 1.20 (2.10/1.75) | **16: Chest trauma and other diseases of the pleura**<br>**N = 1151 (1.0%)**<br>**MR = 3.42 (2.79/0.82)** | S20-S29 (81.1%): 4.25 (2.79/0.66)<br>J90-J94 (18.9%): 1.79 (2.79/1.56) |

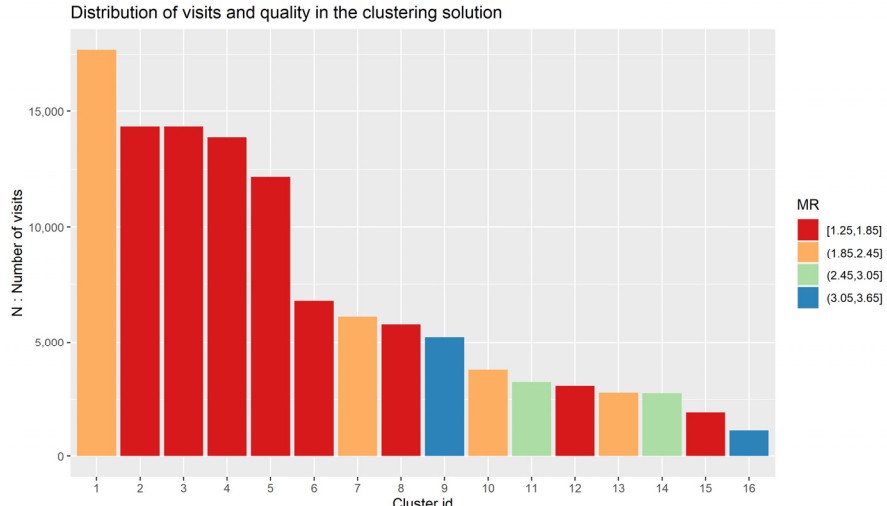

**Figure 2.** Histogram of the distribution of visits and quality in the 16-cluster solution. The MR indicator is illustrated using a 4-color gradient with red color signaling a smaller value.

As illustrated in Table 2 and Figure 2, the clustering solution obtained has a non-uniform spread in terms of visit population, with the frequency of visits gradually decreasing from 15.5% to 1.0%. Indeed, out of the 16 clusters found, we can distinguish a first group with the five biggest clusters, which have similar numbers of visits ranging from 17,705 (15.5%) to 12,139 (10.6%), accounting for 72,108 (63.0%) common visits of the total 114,391 visits; and a second group, with the 11 others, which have several visits steadily decreasing from 6777 (5.9%) to 1151 (1.0%), accounting for 42,509 (37.1%) common visits. Out of the 917 (0.8%) visits with multiple diagnoses, 624 (0.5%) were classified in two clusters and one in three clusters, hence the association between visits and clusters is close to bijective.

### 3.3. Analysis of the Clustering Characteristics

This section aims to illustrate the quality of clustering by analyzing the contents of the clusters obtained. As illustrated in Table 3, the segmentation produced by the clustering enabled analysis of the structuration of the ED health profiles. The number of visits per patient was associated with the cluster size with a number between 1.17 and 1.34 for the first group (cluster 1 to 5) and between 1.04 and 1.18 for the second group (cluster 6 to 16). Given the general aspect of the population and the absence of stratification, each of the 16 clusters was characterized with a wide range of population with standard deviations between 20 and 30 years. However, some clusters have distinct sociodemographic and clinical characteristics that can be summarized with:

- Cluster 1, as suggest the type of health problem it contains, is characterized by a majority of women (71%), a high rate of biological examinations at 64.6% (rank 2nd), a relatively high use of the long triage circuit with a ratio of 1.17 (rank 4th), and a high hospitalization rate of 21% (rank 6th).
- Clusters 2, 4 and 8 have the highest mean age at 49.58 (3rd), 57.86 (1st) and 55.53 (2nd) years, and are associated with the most care-intensive patients. Indeed, they are characterized by the highest rate of hospitalization, at 34% (3rd), 37% (2nd) and 44% (1st) respectively, the highest use of the long triage circuit with ratios of 0.61 (2nd), 0.24 (1st) and 1.12 (3rd) respectively, the highest probability of radiological examination at 55.3% (2nd), 74.5% (1st) and 39.7% (3rd), the highest probability of biological examination, at 66.6% (2nd), 82.2% (1st) and 60.3% (4th), and the highest mean number of medications (prescribed) at 2.08 (2nd), 2.96 (1st), and 2.46 (3rd), respectively.
- Cluster 3 is characterized by the youngest population, with a mean age of 20.84 years, further analyses show that it contains 51.2% of visits with children of 5 years or less, and it is related to winter epidemics, with 23.4% of visits occurring in December and January (compared to the expected 16.6%).
- Clusters from the second group (6 to 16) are, except for cluster 10, characterized by a male predominance, varying between 52% and 64%, with clusters 6, 7 and 16 being the three most concerned with prevalence of men of 61%, 62% and 64% respectively. This is likely associated with the fact that most of these clusters are either directly related to trauma, for seven clusters: 6, 7, 8, 9, 12, 15 and 16, or indirectly for the other four clusters (clusters 10, 11, 13, 14).

### 3.4. Sensitivity Analysis

A sensitivity analysis on a ten times shorter period, that is a nine-week period going from 2 December 2018 to 27 January 2019 during the winter epidemics, was performed to show the robustness of the algorithm regarding the duration of design period. A total of 9796 visits were depicted and filtered to 9317 after excluding 66 out of the 162 blocks (leading to 96 blocks) that had number of occurrences below 10. The HAC-cooc rule (b) approach found an optimal MR of 2.37 (1.72/0.73) at 26 clusters indicating clusters with better quality but with less aggregation than the 2-years period. Using the same number of clusters, the solution at 16 clusters had an MR of 2.04 (1.07/0.525), which is a little bit better than 1.81 (1.40/0.77).

Table 3. Sociodemographic and clinical characteristics of the 16 clusters.

| Cluster Name | Visits, n (%) | Age, Mean | Females Ratio, % | Number of Visits per Patient, Mean | Number of Medications, Mean | Biological Exams Probability (%) | Radiological Exams Probability (%) | Ratio of Short Circuit to Long Circuit Use | Number Admitted to Hospital, n (%) |
|---|---|---|---|---|---|---|---|---|---|
| 1: Digestive disorders, pregnancy, menstruation | 17,705 (15%) | 36.57 | 71% | 1.34 | 1.76 | 64.6% | 34.5% | 1.17 | 3800 (21%) |
| 2: General symptoms and mental disorders | 14,336 (13%) | 49.58 | 53% | 1.17 | 2.08 | 66.6% | 55.3% | 0.61 | 4810 (34%) |
| 3: Infectious diseases | 14,336 (13%) | 20.84 | 49% | 1.28 | 0.97 | 26.2% | 15.4% | 3.94 | 1781 (12%) |
| 4: General symptoms of chronic conditions | 13,893 (12%) | 57.86 | 49% | 1.24 | 2.96 | 82.2% | 74.5% | 0.24 | 6149 (44%) |
| 5: Mental disorders and at-risk behaviours | 12,139 (11%) | 38.63 | 50% | 1.27 | 0.91 | 36.4% | 27.8% | 1.97 | 2693 (22%) |
| 6: Wrist and hand trauma | 6777 (6%) | 32.23 | 38% | 1.07 | 0.35 | 5.6% | 5.0% | 135.55 | 372 (5%) |
| 7 : Head Trauma | 6112 (5%) | 32.26 | 39% | 1.06 | 0.56 | 12.0% | 11.3% | 9.36 | 526 (9%) |
| 8: Hip related trauma and disorders | 5781 (5%) | 55.33 | 40% | 1.18 | 2.46 | 60.3% | 39.7% | 1.12 | 2119 (37%) |
| 9: Feet Trauma | 5224 (5%) | 29.80 | 48% | 1.08 | 0.25 | 4.0% | 3.7% | 66.84 | 137 (3%) |
| 10: Back and Spine disorders | 3777 (3%) | 45.58 | 55% | 1.09 | 1.79 | 33.3% | 24.4% | 2.87 | 682 (18%) |
| 11: Occulomotor disorders | 3235 (3%) | 39.43 | 39% | 1.06 | 0.44 | 7.5% | 6.0% | 13.95 | 152 (5%) |
| 12: Lower limb trauma | 3074 (3%) | 36.18 | 43% | 1.05 | 0.62 | 11.2% | 10.2% | 35.93 | 323 (11%) |
| 13: Cutaneous infections, wounds, and skin disorders | 2787 (2%) | 34.72 | 48% | 1.17 | 0.78 | 26.9% | 9.9% | 9.04 | 414 (15%) |
| 14: Arthropathies | 2760 (2%) | 45.84 | 48% | 1.05 | 1.22 | 29.1% | 25.0% | 3.58 | 473 (17%) |
| 15: Shoulder and arm trauma | 1930 (2%) | 40.96 | 43% | 1.05 | 1.08 | 17.4% | 16.8% | 22.68 | 305 (16%) |
| 16: Chest trauma and other diseases of the pleura | 1151 (1%) | 48.98 | 36% | 1.04 | 1.71 | 38.1% | 37.5% | 2.17 | 303 (26%) |

## 4. Discussion

The multimorbidity clustering algorithm presented in this study provides an original and innovative method to detect comorbidity patterns and describe their structure on a global population scale with a 2-step clustering approach. It enables a coherent and clinically meaningful evaluation of an ED population with diagnosis clusters that embed highly coherent diagnoses within one cluster, and sufficiently differentiate between clusters. From the four statistical techniques of multimorbidity patterns analysis identified by Busija et al. [17], the approach proposed in this study is based on a cluster analysis of diseases with hard clusters, which implies a discrete association between diseases and clusters. In the literature, methods based on this statistical technique used the HAC algorithm with a Jaccard coefficient or Yule's Q as a similarity measure. Using new quality metrics, the present method is compared to the HAC approach with the Jaccard coefficient and Yule's Q coefficient, and with an MCA approach, usually used for clustering people but adjustable for clustering diagnoses. As such, compared to the state-of-the-art HAC methods, the present approach proposes three innovations: a new similarity measure based on the relative risk of co-occurrence, an adaptation of the HAC algorithm to this type of statistical context, which is neither categorical nor numerical, and the introduction of quality metrics based on the new similarity measure. Furthermore, the approach is used in a novel context with the analysis of a global ED population and its specific multimorbidity patterns.

The clustering strategy can be compared to other approaches found in the literature based on its three stages of design, necessary to any multimorbidity clustering study, namely the stage of choosing the representation and selecting both diagnosis and population; the stage of choosing an embedded space and a similarity measure to compare entities; and the final stage of choosing the clustering algorithm and the appropriate quality indicators. The comparison of multimorbidity analysis methods, which is still rare [36], will help explain why the other approaches tested, named here HAC Jaccard and MCA HAC and similar to those found in the literature [20,21], failed to provide adequate results in the present context.

The choice of representation and the selection of the diagnoses and population depend primarily on the end goal of the clustering study, and secondarily on the aim of limiting sampling biases in the co-occurrence count of the diagnoses. Since the present study was further used to analyze the evolution of ED visits over the 2-year study period, and since it requires an assessment of the general population, we included all diagnoses that had an occurrence of at least 10 episodes of visits, and all visits with at least one of these diagnoses. We also did not apply any stratification, and we chose a high block level representation of the ICD10 classification, as it provides adequate semantic precision for a general population assessment, limiting the number of clusters and the complexity of the diagnosis relationships. Hybrid approaches using both ICD10 concept similarity and semantic and co-occurrence similarity could be conceptualized, but would represent additional challenges to find a coherent measure that would strike a balance between the two heterogeneous similarities [9–11]. The exclusion of specific diagnoses, like R codes (*symptoms, signs, and abnormal clinical and laboratory findings, not elsewhere classified*) and Z codes (*factors influencing health status and contact with health services*) [21], could have enhanced the quality of the clustering, as they are linked to many heterogeneous health problems, but would have ignored 28.9% of all ED visits. Other approaches, identified by Prados-Torres [16], used diagnoses from the International Classification of Primary Care (ICPC), the ICD-9 and ICD-10 classification and others from an ad hoc list, limiting the number of baseline diagnoses between 10 to 50 for most of them (13 out of 14), by focusing on a specific set of diseases or by collapsing diagnoses into groups. Eight out of the 14 studies focused on elderly patients aged over 60 years, three used stratification by age and sex, and 10 studies selected the diagnosis based on their chronicity. These tendencies are further confirmed in the literature review by Busija et al. [17], where 46 of the 51 studies identified used 32 or fewer baseline diagnoses, and 32 studies focussed on populations older than 50 years. Therefore, with 151 blocks of diagnoses and a population spanning all

ages, the present study proposes one of the most general multimorbidity approaches, with readable and coherent results through the 16 clusters found.

The choice of similarity measure is the most crucial element of the clustering algorithm, as it will dictate how the entities are clustered together. In the context of categorical variables, and furthermore with the specificities of the diagnosis variable presented here, the design of the measure requires specific attention [29]. Particularly, with each diagnostic label represented as a binary variable, multimorbidity pattern analysis relies on statistical co-occurrences in patients of pairs of diagnoses. These co-occurrences are either measured during one visit or at another time, during multiple visits, as it is the case in the present study and in most other studies [16,17]. This timeframe raises the problem of the design choices regarding the duration between co-occurrences. The design should consider whether the co-occurrences are meaningful, given the time interval between them, the population observed, and the study period. The present study empirically limited this duration to 6 months out of a 2-year study period, as this seemed to be a meaningful design for ED visits. Integrating the temporal aspect of co-occurrences directly into the proximity measure could also be considered, but would make the clustering design and interpretation of the results more complex. The choice of study period duration will also have an impact the quality of the results. Indeed, while being more time specific, a shorter period provides less occurrences and co-occurrences and therefore statistical sampling than a longer one. The sensitivity analysis on a nine-week period shows that the algorithm is quite robust regarding the study-period for the construction of a performant solution. However, with more than a third of all the blocks being filtered due to a number of occurrences less than 10, it also shows that the length of the study period is crucial to have enough statistical sampling of the occurrences and co-occurrences. Thus, shorter periods would begin to obscure the multimorbidity patterns that can be found in the ED population.

The diagnoses co-occurrences can be expressed with contingency tables, with positive matches, negative matches, and positive-negative matches between diagnoses. From these tables, three proximity measures are often used in the literature [16,17]: the Jaccard coefficient, that relies on positive matches to measure the tetrachoric correlation matrix, and Yule Q that relies on all types of matches. However, negative and negative-positive matches are very poorly informative on the diseases' relationships compared to positive matches, because they can be mainly attributed to the multimorbidity context, i.e., the number of diagnoses per patient. This is constrained by their number of visits, which do not represent an exhaustive view of the patient's health problems. Therefore, especially in the low multimorbidity context of general populations, the RR measure presented in this study and the Jaccard measure are appropriate measures based on positive matches. However, as shown in the results section, the Jaccard measure is not efficient to produce good results with the proposed quality indicators here. Furthermore, the RR measure takes better account of the marginal propensities of each diagnosis and it has a more profound probabilistic interpretation, where the value 1 can be interpreted as probabilistic independence between the occurrences of diagnoses, here on a patient ED episode of visits. Unexpectedly, Yule's Q coefficient provided good results, but is harder to interpret than the RR measure. This suggests that it can also adequately capture the association due to positive matches, although not being focussed on this aspect.

The choice of clustering algorithm leads to the strategy to group data in clusters and the design of the measure of cluster quality. This process depends on the nature of the data, their similarities, their distribution in space, if it has one, and on the nature of the desired cluster. Most traditional algorithms expect data to be in a Euclidean space with a distance function, respecting the triangle inequality between three elements $(d(a,c) \leq d(a,b) + d(b,c))$. With categorical data, this assumption is not verified and the use of pseudo-distance [29], or the embedding of the categorical data in a Euclidean space, is possible, but is not appropriate for the statistical comparison of diagnoses for two main reasons. Firstly, the interpretation of clustering quality from the use of pseudo-distance is quite difficult and generally considers each diagnostic label entity with the

same importance for the clustering, whereas their occurrences in the data can be drastically different. Secondly, the embedding of diagnoses in a Euclidean space would result in a loss of information because the statistical relationship between two diagnoses suggests, but does not imply, any relationship with others diagnoses, contrary to a distance link, which would bound the distance with other entities based on the triangle inequality. Embedding diagnoses with Multiple Correspondence Analysis is therefore inappropriate. Furthermore, it does not produce valuable results in our context and was designed initially to analyze surveys [37], which do not capture the specificity of diagnosis variables with the importance of positive matches.

The design of the clustering algorithm also depends on the use of valid quality indicators and on the choice of type of clusters. Our design of a non-distance-based HAC approach is the first to propose a dedicated and coherent quality indicator for this type of approach. Indeed, contrary to numerical and categorical clustering, no usual quality metrics currently exist to measure the compactness and separation of this type of cluster. Comparable studies rely on an empirical inspection of the dendrogram to determine the number of clusters, without describing the statistical quality of the cluster found, e.g., Cornell et al. who used clinical criteria [20]. A methodology such as the one used here with the RR measure could be used to construct quality metrics from the Jaccard coefficient or Yule's Q, but this is beyond the scope of this study. Other methodologies from network clustering could be a source of inspiration for constructing new cost functions like MR. Particularly, community scoring functions combining internal and external connectivity, such as conductance [38], could be adapted for graphs with weighted nodes and weighted links like the underlying graph of the present study with its diagnoses occurrences and co-occurrences used as weights. The membership ratio MR, constructed here for multimorbidity pattern analysis, can be analyzed at several levels: here at block, cluster, and global level. As such, it has proven to be an efficient way to respond to the problem of clustering diagnostic entities in a meaningful way. The hard nature of the constructed cluster, where each diagnosis belongs univocally to only one cluster, was chosen here to facilitate the interpretation of the cluster and of the evolution of ED visits. Fuzzy clusters, produced by exploratory factor analysis or by fuzzy c-means for example, where diagnoses belong to multiple clusters/factors using weighted links, seems more appropriate for disease analysis and the impact of comorbidities on risk of undesired health outcomes, as it can discover more complex patterns [19,39].

Due to the novelty of the ED global population context, the clusters obtained can hardly be compared with those found in the literature, which focuses almost entirely on old, chronically ill patients [16,17]. The only two comparable clusters are those of cluster 2 (General symptoms and mental disorders) and cluster 4 (General Symptoms and chronic conditions), which can, respectively, be assimilated to the mental health conditions grouping, and the cardio-metabolic conditions grouping found by Busija et al. [17]. This can be explained by the fact that these clusters contain the oldest, most chronic populations, making them coherent with the population in the literature. As shown by the results, these populations are also the most care-intensive, accounting for 25% of the visits according to cluster 2 and 4 prevalence. Cluster 1 (Digestive disorders, pregnancy, menstruation), cluster 3 (Infectious diseases) and cluster 5 (Mental disorders and at-risk behaviors) represent three new categories of multimorbidity clusters that characterize the specific health problems of the ED encountered in populations other than the elderly and chronically ill. With 1.17 to 1.34 visits per patient, these clusters are the most representative of the health problems with readmission behaviors encountered in the ED. This contrasts with clusters 6 to 16, which are all trauma-related and much less likely to result in readmission patterns, with 1.05 to 1.18 visits per patient. This can be linked to the fact that these types of health problems are most often acute, and not chronic. Indeed, they require unscheduled care provided essentially by EDs, and are hard to prevent by healthcare consultations, since they are often associated with accidental or at-risk behaviors.

## 5. Conclusions

Multimorbidity pattern analysis is an issue that has been widely studied in the epidemiology literature. With the widespread use of diagnoses to summarize the health problem of each patient and the need to assess their complexity, as well as the associated risk, it is crucial to develop pertinent methodologies that consider the specific properties of diagnosis variables. The present study proposes an original approach to cluster diagnoses or diseases based on their statistical co-occurrences. It can be easily exported to many contexts, as it relies only on the use of a diagnoses system and on co-occurrence pattern entities. It has shown to be efficacious within the low morbidity context of the general ED population, but could also be used in the classical context of chronic diseases and could inspire new research in the field.

Further research should focus on the optimization of the proposed quality indicators and on the modelling of the similarities. This could lead to the initiation of better method practices regarding the performance analysis of clustering that should enable a better statistical comparison between studies. We will also validate the proposed approach in future studies by applying it to more classic chronic diseases contexts with other datasets such as the French Health Data Hub [40].

**Supplementary Materials:** The following are available online at https://www.mdpi.com/2076-3417/11/5/2373/s1, Table S1: Full diagnostic content of the 16 clusters and cluster quality, Table S2: Labels of the ICD10 Blocks found in the 16 clusters

**Author Contributions:** A.W., F.M.-C., F.Y., J.C., D.L. and S.S. designed the study. S.S., D.L., contributed to the acquisition of data. A.W. and J.C. conducted the statistical analysis. A.W., F.M.-C., F.Y., J.C., D.L. and S.S. contributed to drafting the article or revising it critically for important intellectual content. All authors have read and agreed to the published version of the manuscript.

**Funding:** This research received no external funding

**Institutional Review Board Statement:** This study was performed in compliance with national legislation regarding epidemiological studies (declaration N°2203674v0, dated 24/07/2018). Since the study was wholly observational and used only anonymized data, neither ethics approval nor a specific written informed consent from participants was required in France for this retrospective database study. In accordance with French ethical directives, the requirement for written informed consent was waived because the study was strictly observational, and all data were blinded. (ref: French Public Health Code. Article R. 1121-2. [http://www.legifrance.gouv.fr] (accessed on 28 February 2021)). According to the French Public Health Code, this research did not require an ethical committee (ref: French Public Health Code. Article R. 1121-2. [http://www.legifrance.gouv.fr] (accessed on 28 February 2021)). The study is conducted according to the legislation for medical information: it was declared with the national registry of health research under the number N°1113130319. Patients were informed that the study was being carried out through the hospital's registry of ongoing studies.

**Informed Consent Statement:** Not applicable.

**Data Availability Statement:** The main dataset containing the patient's exact stay with arrival date, ID number, stay number, age, gender, diagnoses, and other information described in the methods section cannot be made publicly available due to its confidential nature. Access is restricted by law to researchers that have complied with national legislation regarding epidemiological studies. It can be made available from the corresponding author on reasonable and legal request.

**Acknowledgments:** We would like to thank Fiona Escarnot for her contribution to the linguistic revision of this paper.

**Conflicts of Interest:** The authors declare no conflict of interest.

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
