# Peer review of "Clustering of a Health Dataset Using Diagnosis Co-Occurrences"

_applsci, doi:10.3390/app11052373_

Round 1

Reviewer 1 Report

Dear authors,

you present here an unsupervised approach for grouping patient populations based on a series of characteristics, introducing also a smart manner of avoiding categorical data but still getting information by their underlying presence. Thus, you claim that a new clustering that is based on Multimorbidity Patterns and exploits a well-defined metric can address this kind of problem. Although it seems that you have studied the literature, and you have in-depth knowledge of the need of this kind of problem, there are some important issues regarding your manuscript. I mention them here, and I would expect a revision of this work which must have tackled these defects, because in the current format your work is not close to a scientific manuscript that would attract readers' interest.

  1. All the formulas and the typos into the manuscript are distorted, putting obstacles to the reader's ability. Why does this happen across all the manuscript? 
  2. Same problems in algorithm 1. I cannot read it, and I should not visit the supplementary material for getting access to your algorithm. You have to structure him into the manuscript.
  3. Table 1 and figure 2 are not properly presented. You have to find a manner of presenting them so as to be readily available to the reader, without the need of scrolling on 3 separate pages for getting this kind of information. Use some explanatory graphs and stats for presenting the actual context of table 1, and of course re-structure figure 2 since it contains too much and specific indicators. 
  4. Use bullets for presenting the findings of subsection 3.3
  5. shorten your sentences across the manuscript, trying to highlight the most important findings or other statements that hold in the specific problem that you examine
  6. it is important to make some comparisons of the proposed clustering algorithm with other similar algorithms. Although they do not use the same criterion with yours, you have to compute usual clustering metrics with at least 3 other approaches, and present us the characteristics of their clusters, so as to understand better the actual contribution and the assets of your work. Do other clustering algorithms also reach to similar groupings or do their clusters present similar consistency regarding similar or other criteria?
  7. Moreover, I would expect to perform the same clustering but with data from shorter periods. Does your approach present similar quality regarding its discriminative ability?
  8. A complexity analysis of your algorithm should also benefit your total work.

Reviewer 2 Report

The authors propose a hierarchical agglomerative clustering algorithm based on multimorbidity analysis for clustering health dataset using diagnosis co-occurrences.

The proposed study is interesting but there are some points that the authors should better discuss.

The authors should be better described the novelties of their study with respect to existing ones. In particular, the author should discuss limitation and cons of the examined approaches. Furthermore, the authors should provide more details and discussion about the obtained results. The Discussion section also needs to be improved by analyzing the outcome of evaluation section.

I suggest to further analyze more recent approaches about the examined topics. In particular, I suggest the following papers to further investigate clustering method based on game theory and emotional personality and mood in clustering process in the introduction section:

1) Evolutionary game theoretical on-line event detection over tweet streams. Knowledge-Based Systems 211: 106563.

2) An emotional recommender system for music. IEEE Intelligent Systems.

Finally, I suggest to perform a linguistic revision.

Round 2

Reviewer 1 Report

Dear authors,

your answers and the modifications that you have applied improved the total manuscript. I understand the reasons why extensive comparisons with other clustering algorithms is not feasible in-depth. I mention some minor changes which could further improve your work:

i) remove MR from Table 1 from the table, and just mention it on the header.

ii) i would avoid depicting all the recorded variables in the main text, showing these with the largest prevalence and providing some stats about the long-tail distribution. Consider it again.

iii) Avoid the specific colors in Fig. 2 (these gradients are not well discriminated).

iv) Please add some references regarding the future work and extend the last Section

v) Elaborate formulas, especially 6 and 7 which are too close

vi) you comment my previous question about the use of less data. I catch your point, but i believe that a comparison with using a subsample of the total data could depict some useful information about the robustness of the algorithm or its bias regarding the limited input instances.

Your work is ready for publication, but do not lose the chance to still improve it.

Reviewer 2 Report

I think that the authors have addressed all my concerns. 

Author Response

We thank the reviewer for his recommendations.